



**1   Liquid-liquid phase separation in particles containing secondary organic**

**2   material free of inorganic salts**

Mijung Song[1,2], Pengfei Liu[3], Scot T. Martin[3,4], Allan K. Bertram[2*]
[1] {Department of Earth and Environmental Sciences, Chonbuk National University, Jeollabuk-
do, Republic of Korea}
[2] {Department of Chemistry, University of British Columbia, Vancouver, BC, V6T 1Z1, Canada}
[3] {John A. Paulson School of Engineering and Applied Sciences, Harvard University,
Cambridge, Massachusetts 02138, USA}
[4] {Department of Earth and Planetary Sciences, Harvard University, Cambridge, Massachusetts
02138, USA}
Correspondence to: A. K. Bertram (bertram@chem.ubc.ca)
**Abstract**
Particles containing secondary organic material (SOM) are ubiquitous in the atmosphere and play
a role in climate and air quality. Recently, research has shown that liquid-liquid phase separation
(LLPS) occurs at high relative humidities (RH) (greater than ~ 95 %) in α-pinene-derived SOM
particles free of inorganic salts while LLPS does not occur in isoprene-derived SOM particles free
of inorganic salts. We expand on these findings by investigating LLPS in SOM particles free of
inorganic salts produced from ozonolysis of β-caryophyllene, ozonolysis of limonene, and photo-
oxidation of toluene. LLPS was observed at greater than ~95 % RH in the biogenic SOM particles
derived from β-caryophyllene and limonene while LLPS was not observed in the anthropogenic
SOM particles derived from toluene at 290 ± 1 K. This work combined with the earlier work on
LLPS in SOM particles free of inorganic salts suggests that the occurrence of LLPS in SOM
particles free of inorganic salts is related to the average oxygen-to-carbon elemental ratio (O:C) of
the organic material.  When the average O:C is between 0.25 and 0.60, LLPS was observed, but
when the average O:C was between 0.52 and 1.3, LLPS was not observed. These results help





explain the difference between the hygroscopic parameter $\kappa$ of SOM particles measured above and
below water saturation in the laboratory and field, and have implications for predicting the cloud
condensation nucleation properties of SOM particles.
**1 Introduction**
Secondary organic material (SOM) is produced in the atmosphere by the oxidation of volatile
organic compounds (VOCs) such as α-pinene and isoprene from trees and toluene from
anthropogenic sources. Once formed, the low volatility oxidation products can partition to the
particle phase to form SOM containing particles (Hallquist et al., 2009; Ervens et al., 2011). SOM
accounts for approximately 20 – 80 % of the submicrometer particle mass in the atmosphere
(Zhang et al., 2007; Jimenez et al., 2009). Although the exact chemical composition of SOM in
atmospheric particles remains an active area of research, laboratory and field studies have shown
that the average oxygen-to-carbon elemental ratio (O:C) of these particles ranges from 0.2 to 1.0
(Chen et al., 2009; Jimenez et al., 2009; Heald et al., 2010; Takahama et al., 2011).
As the relative humidity (RH) varies in the atmosphere, SOM containing particles can undergo
several different phase transitions with implications for the cloud condensation nuclei (CCN)
properties, optical properties, reactivity, and growth of these particles (Martin et al., 2000;
Raymond and Pandis, 2002; Bilde and Svenningsson, 2004; Zuend et al., 2010; Kuwata and Martin,
2012; Brunamonti et al., 2015). One possible phase transition that SOM particles may undergo as
RH varies in the atmosphere is liquid-liquid phase separation (LLPS) (Pankow, 2003; Marcolli et
al., 2006; Ciobanu et al., 2009: Zuend and Seinfeld, 2012; Veghte et al., 2013; O'Brien et al.,
2015). LLPS in particles containing both SOM and inorganic salts has been the focus of many
recent studies, and it is now established that SOM particles mixed with inorganic salts can undergo
LLPS in the atmosphere when the O:C of the organic material is roughly less than 0.8 (Bertram et
al., 2011; Krieger et al., 2012; Smith et al., 2012; Song et al., 2012a; Schill and Tolbert, 2013;
Song et al., 2013; You et al., 2013; You et al., 2014).
Recently, researchers have also focused on LLPS in SOM particles free of inorganic salts. Peters
et al. (2006) suggested that a miscibility gap in particles containing organic polymers at high RH
may lead to a non-classical pathway for CCN activation. Renbaum-Wolff et al. (2016) showed that
α-pinene-derived SOM free of inorganic salts can undergo LLPS at high RH values (~95 to 100 %)



resulting in altered CCN properties. In addition, they showed that LLPS in SOM particles will lead to a different hygroscopic parameter, $\kappa$, at subsaturated conditions compared to supersaturated conditions. The implication is that the CCN activity of SOM particles, if they undergo LLPS, is higher than predicted from subsaturated hygroscopicity measurements. Most recently, Rastak et al. (2017) observed that isoprene-derived SOM particles do not undergo LLPS even at high RH. Rastak et al. (2017) also used these results together with thermodynamic calculations to explain the hygroscopic properties of biogenic organic aerosol particles in the laboratory and the field.

Here we expand on the studies by Renbaum-Wolff et al. (2016) and Rastak et al. (2017) by investigated LLPS in SOM particles generated by the ozonolysis of limonene, ozonolysis of β-caryophyllene, and photo-oxidation of toluene. Limonene and β-caryophyllene are both biogenic VOCs, while toluene is an anthropogenic VOC (Kanakidou et al., 2005). Both limonene-derived and β-caryophyllene-derived SOM particles have been used as proxies of biogenic SOM particles in the atmosphere (Bateman et al., 2009; Alfarra et al., 2012; Kundu et al., 2012; Frosch et al., 2013; Liu et al., 2013), while toluene-derived SOM has been used as a proxy for anthropogenic SOM particles (Pandis et al., 1992; Robinson et al., 2013; Liu et al., 2016; Song et al., 2016; Ye et al., 2016).

## 2 Methods

### 2.1 Production of secondary organic materials

SOM particles were generated via β-caryophyllene ozonolysis and limonene ozonolysis in a flow tube reactor (Table 1) and via toluene photo-oxidation in an oxidation flow reactor (OFR) (Table 2). The method of particle generation in the flow tube reactor was described in Shrestha et al. (2013) and the methods of particle generation in the OFR (Kang et al., 2007) were given in Liu et al. (2015). The flow tube reactor was operated at a flow rate of 3.5 L min$^{-1}$ (with a residence time of 38 s) and < 5% RH. The OFR was operated at flow rates of 7.0 to 9.5 L min$^{-1}$ (with a residence time of 80 to 110 s) and 13 ± 3 % RH. Both reactors were operated at a temperature of 293 ± 2 K.

Table 1 lists the experimental conditions for the production of SOM via ozonolysis. For the particle generation via ozonolysis, ozone was produced by irradiating pure air (Aadco 737 Pure Air Generator) with ultraviolet emission from a mercury lamp (λ = 185 nm). Ozone concentrations





used for ozonolysis ranged from 12 - 30 ppm for β-caryophyllene and 13 - 30 ppm for limonene
(Table 1). β-caryophyllene and limonene (Sigma Aldrich, ≥ 99 %) were dissolved in 2-butanol
(Sigma-Aldrich, ≥ 99.5 %). These organic solutions were injected into a glass round-bottom flask
held at 310 K, where the organic liquids vaporized at the tip of a syringe. The organic vapor was
then swept into the reactor where ozonolysis took place to form SOM and particles. The injected
precursor concentrations were 0.03 - 0.7 ppm for β-caryophyllene and 0.07 - 2.0 ppm for limonene
in the main flow of the reactor. In the ozonolysis experiments, butanol served as an OH radical
scavenger.
Table 2 presents the experimental conditions for the production of SOM via photo-oxidation. For
the particle generation via photo-oxidation, hydroxyl radicals were produced in the OFR by the
photochemical reactions:
$O_3 + h\nu \rightarrow O_2 + O(^1D)$,                    (R1)
$O(^1D) + H_2O \rightarrow 2OH$                    (R2)
Ozone was again produced by irradiating pure air (Aadco 737 Pure Air Generator) with ultraviolet
emission from a mercury lamp (λ = 185 nm). Ozone concentrations used in the photo-oxidation
studies were 30 ppm (Table 1). Toluene (Sigma-Aldrich, 99 %) was injected and vaporized in a
flask, and the vapors were swept into the OFR by purified air. The injected toluene concentrations
were 0.2 - 1.0 ppm.
**2.2 Production of supermicron SOM particles on hydrophobic substrates**
At the outlet of the flow tube reactor and OFR, the sub-micrometer SOM particles were collected
on hydrophobic surfaces. The limonene-derived SOM and toluene-derived SOM particles were
collected onto glass slides coated with trichloro(*1H,1H,2H,2H*-perfluorooctyl)silane (Sigma-
Aldrich, 97%). The coating procedure is described in Knopf (2003). The β-caryophyllene-derived
SOM particles were collected onto teflon substrates.




Two different methods were used to collect submicron particles on hydrophobic substrates (see
Tables 1 and 2). The first method used was an electrostatic precipitator (TSI 3089, USA). In this
case, the resulting SOM particles on the hydrophobic substrates were smaller than ~10 µm. Since
particles 20 - 80 µm in diameter were required for the LLPS experiments, the following method
was used to coagulate the sub-10 µm particles into 20 - 80 µm particles: first the substrate
containing the SOM particles was placed in a RH-controlled flow-cell (Parsons et al., 2004; Pant
et al., 2006; Song et al., 2012b). The RH in the flow-cell was then set to over 100 % for 30 - 60
min to grow and coagulate the SOM particles. The RH in the flow-cell was then decreased to ~80
- 90 % RH to evaporate the water. During the experiments, the particles were observed using a
reflectance microscope (Zeiss Axiotech, 50×). These growth and coagulation processes resulted in
SOM particles consisting of 20 - 80 µm in diameter (Song et al., 2015; Renbaum-Wolff et al.,

12   2016).

In the second method used to collect SOM particles collected on a hydrophobic substrate, a single
stage impactor was used (Prenni et al., 2009; Pöschl et al., 2010; Hosny et al., 2016). In this case,
the SOM particles after collection were as big as 100 µm due to coagulation during the collection
process. Since the particles were already large enough for the LLPS experiments, they were used
directly without the need for the growth and coagulation experiments described above. Both
methods used to collect SOM particles collected both the water-soluble and water-insoluble
components of the SOM particles.
**2.3 Optical microscopy of supermicron SOM particles**
For the LLPS experiments, the hydrophobic substrate containing SOM particles with sizes in the
range of 20 to 80 µm in diameters was mounted in a temperature and RH controlled flow-cell
coupled to an optical reflectance microscope (Zess Axiotech, 50× objective) (Parsons et al., 2004;
Pant et al., 2006; Song et al., 2012b). The temperature of the cell was 290 ± 1 K in all experiments.
RH in the cell was regulated by varying the ratio of a dry and humidified $N_2$ flow. The total flow
rate was ~1200 sccm. The RH was measured using a hygrometer with a chilled mirror sensor
(General Eastern, Canada), which was calibrated using the deliquescence RH for pure ammonium
sulfate particles (80% RH at 293 K, Martin, 2000). After calibration, the uncertainty of the





hygrometer was ± 2.0 % RH. At the beginning of LLPS experiments the SOM particles were
equilibrated at ~100 % RH for 15 minutes. Then the RH was reduced from ~100 to ~0 % RH at a
rate of 0.1 to 0.5 % RH min⁻¹, and subsequently increased to ~100 % RH at a rate of 0.1 to 0.5 %
RH min⁻¹. During the humidity cycle, optical images of the SOM particles were recorded every 5
- 10 seconds using a CCD camera.
**3 Results and Discussion**
**3.1 β-caryophyllene-derived and limonene-derived SOM particles**
Humidity cycles at $290 \pm 1$ K were performed for β-caryophyllene-derived SOM particles
generated with mass concentrations of 15 - 4000 µg m⁻³ and limonene-derived SOM generated
with mass concentrations of 80 - 7000 µg m⁻³ (Table 1). In all cases, LLPS was observed at high
RH. Table 1 summarizes the results during humidity cycles. Shown in Fig. 1a and Movie S1
(Supplementary Material) are examples of optical images of a β-caryophyllene-derived SOM
particle as a function of increasing RH for the particle mass concentrations of 2000 - 4000 µg m⁻³.
Shown in Fig. 1b and Movie S2 (Supplementary Material) are examples of optical images of a
limonene-derived SOM particle as a function of increasing RH for the particle mass concentrations
of 7000 µg m⁻³. For both types of SOM particles, only one phase was observed for RH values from
0 to ~ 90 %. Note, the light-colored circle in the center of the particles at 90.5 % RH for β-
caryophyllene-derived SOM and at 95.0 % RH for limonene-derived SOM is an optical effect due
to the light scattering from a hemispherical particle (Bertram et al., 2011). In Fig. 1, LLPS is
observed at 91.5 % RH for the β-caryophyllene-derived SOM particle and at 95.3 % RH for the
limonene-derived SOM particle. LLPS began with the formation of many small inclusions of a
second phase, and in both cases the phase transition occurred over a narrow range of RH. The
appearance of many small inclusions is consistent with the phase transition occurring by spinodal
decomposition (Ciobanu et al., 2009). Renbaum-Wolff et al. (2016) also observed LLPS in SOM
particles produced from α-pinene ozonolysis by spinodal decomposition. The small inclusions
coagulated to larger droplets in the β-caryophyllene-derived SOM at 92.5 % RH and in the
limonene-derived SOM at 96.1 % RH (Figs. 1a and 1b). The new phase formed is a water-rich
phase while the other phase is an SOM-rich phase (Renbaum-Wolff et al., 2016). At the highest
RH investigated, the majority of the water-rich phase was in the center of the particles. Such core-




shell morphology on a hydrophobic slide glass has been observed previously in dicarboxylic
acids/ammonium sulfate/H₂O particles by Song et al. (2012b). After formation of the core-shell
morphology consisting of inner and outer phase, the two liquid phases co-existed as high as ~100 %
RH. Upon drying, the two liquid phases merge into one liquid phase. This merging process
occurred at 90.9 % RH for β-caryophyllene-derived SOM and 95.6 % RH for limonene-derived
SOM. Movies of the merging process are shown in the Supplementary Material (Movies S3 and
S4).
To determine whether the occurrence of LLPS depends on the SOM particle mass concentrations
used when generating the SOM, a wide range of the particle mass concentrations covering 15 –
7000 µg m⁻³ were investigated (Table 1). Illustrated in Fig. 2a and 2b is the RH at which two
phases were observed during humidity cycles as a function of the mass concentrations of the β-
caryophyllene-derived and limonene-derived SOM samples. Triangles represent merging relative
humidites (MRH) of two liquid phases upon drying and circles represents separation relative
humidity (SRH) upon moistening.
LLPS was observed at 93.6 ± 1.5 % RH in the β-caryophyllene-derived SOM particles for the
particle mass concentrations of 15 – 4000 µg m⁻³ (Fig. 2a). In the limonene-derived SOM particles,
LLPS occured at 96.1 ± 2.1 % RH for the particle mass concentrations of 80 – 7000 µg m⁻³ (Fig.
2b). LLPS occurred at 96.0 ± 0.7 % RH in α-pinene-derived SOM particles for the mass
concentrations of 75 - 11000 µg m⁻³ (Renbaum-Wolff et al., 2016) (Fig. 2c). As shown in Fig. 2,
the SRH and MRH of the β-caryophyllene-derived SOM, limonene-derived SOM, and α-pinene-
derived SOM particles do not depend strongly on the SOM particle mass concentrations used to
generate the SOM.

## 3.2 Toluene-derived SOM

Humidity cycles were also performed for SOM particles generated from photo-oxidation of
toluene, using particle mass concentrations of 80 - 1000 µg m⁻³ in the reactor (Table 2). None of
the toluene-derived SOM particles underwent LLPS during RH cycling even at high RH (Table 2).
Shown in Fig. 3 and Movie S5 (Supplementary Material) are optical images of toluene-derived
SOM particle for the particle mass concentrations of 80 - 100 µg m⁻³. Images in Fig. 3 and Movie





S5 were recorded as the RH was increased. No LLPS was observed in the SOM particles during
RH cycling between 0 and 100 %. Rastak et al. (2017) did not observed LLPS in isoprene-derived
SOM particles for the mass concentrations of 60 - 1000 µg m$^{-3}$
**3.3 Relation between LLPS and O:C.**
Summarized in Table 3 are the average SRH values determined in our work and by Renbaum-
Wolff et al. (2016) and Rastak et al. (2017). The average SRH values were based on the SRH
values during humidity cycles determined for the different SOM mass concentrations used to
generate the SOM since no strong dependence on mass concentrations was apparent. Also included
in Table 3 is the range of O:C values previously reported in the literature for the studied SOM
particles.  Based on the data shown in Table 3, there appears to be a relationship between the
occurrence of LLPS and the average O:C of the organic material: when the average O:C was
between 0.25 and 0.60, LLPS was observed, but when the average O:C was between 0.52 and 1.3,
LLPS was not observed.  This trend is also apparent in Fig. 4a, where the data in Table 3 is plotted.
The relationship between average O:C and LLPS is consistent with previous studies that explored
the miscibility gap in bulk solutions (see Table 1 in Ganbavale et al., 2015). When the average
O:C of the organic material was low in a system containing two organic components with water,
LLPS was observed. For example, LLPS was observed in a mixture of 1-butanol (O:C = 0.25), 1-
propanol (O:C = 0.20), and water (Gomis-Yagües et al., 1998) and in a mixture of 1-pentanol (O:C
= 0.20), acetone (O:C=0.33), and water (Tiryaki et al., 1994). On the other hand, when the average
O:C of the organic material was high in a system containing two organics and water, LLPS was
not observed. For example, LLPS was not observed in a mixture of acetic acid (O:C=1.00), ethanol
(O:C=0.50), and water (Pickering, 1893).
**4. Implications**
As mentioned in the introduction, Petters et al. (2006), Renbaum-Wolff et al. (2016), and Rastak
et al. (2017) showed using thermodynamic calculations that SOM particles that undergo LLPS at
high RH values have modified CCN properties. Hence, LLPS should be considered when





predicting the CCN properties of SOM particles derived from α-pinene ozonolysis, β-
caryophyllene ozonolysis, and limonene ozonolysis. A caveat is that the mass concentrations used
when generating the SOM particles in our experiments was larger than normally found in the
atmosphere (Zhang et al., 2007; Jimenez et al., 2009; Spracklen et al., 2011; Li et al., 2015).
Additional studies are needed to confirm LLPS in SOM particles generated using more
atmospherically relavant SOM mass concentrations.
Discrepancy between the hygroscopic parameter, $\kappa$, (Petters and Kreidenweis, 2007) measured
below water saturation ($\kappa_{HGF}$) and above water saturation ($\kappa_{CCN}$) in SOM particles have been
reported in several studies (Petters et al., 2006; Prenni et al., 2007; Juranyi et al., 2009; Petters et
al., 2009; Good et al., 2010; Irwin et al., 2010; Massoli et al., 2010; Dusek et al., 2011; Irwin et
al., 2011; Hersey et al., 2013; Pajunoja et al., 2015; Zhao et al., 2016). Petters et al. (2006),
Renbaum-Wolff et al. (2016) and Rastak et al. (2017) suggested that such discrepancies are
expected in systems that undergo LLPS at high RH. Summarized in Table 4 and Fig. 4b is literature
data on the difference between $\kappa_{HGF}$ and $\kappa_{CCN}$ (denoted $\Delta\kappa$) as a function of average O:C of the
organic material (Prenni et al., 2007; Massoli et al., 2010; Pajunoja et al., 2015). Figure 4b suggests
that $\Delta\kappa$ is related to the average O:C of the organic material. Figure 4a and 4b combined suggests
that when the average O:C is small, LLPS occurs and the difference between $\kappa_{HGF}$ and $\kappa_{CCN}$ is
large. On the other hand, when the average O:C is large, LLPS does not occur and the difference
between $\kappa_{HGF}$ and $\kappa_{CCN}$ is small. Figure 4 provides additional support for the suggestion that the
LLPS is related to the discrepancies between $\kappa_{HGF}$ and $\kappa_{CCN}$.
**Acknowledgments**
This work was supported by the Natural Sciences and Engineering Research Council of Canada.
Support from the US National Science Foundation (AGS-1640378) and the US Department of
Energy (DE-SC0012792) is also acknowledged. Mijung Song acknowledges support from a
National Research Foundation of Korea (NRF) grant funded by the Korea Government (MSIP)
(2016R1C1B1009243).





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





Table 1. Experimental conditions for produced and collection of SOM particles by ozonolysis and
measured separation relative humidities (SRH) upon moistening and merging relative humidities
(MRH) upon drying of the collected particles. Particles were collected on hydrophobic substrates
using an electrostatic precipitator or single stage impactor.

| SOM sample | VOC conc. (ppm) | $O_3$ conc. (ppm) | SOM mass conc. ($\mu g\ m^{-3}$) | Flow rate for SOM particle production ($L\ m^{-1}$) | Collection time (hour) | Collection method | MRH (%) | SRH (%) |
|---|---|---|---|---|---|---|---|---|
| β-caryophyllene 1 | 0.03 | 30 | 15-30 | 7.0 | 24 | Single stage impactor | 92.7 | 94.9 |
| β-caryophyllene 2 | 0.03 | 30 | 15-30 | 7.0 | 46 | Single stage impactor | 95.0 | 94.4 |
| β-caryophyllene 3 | 0.7 | 12 | 2000-4000 | 3.5 | 6 | Electrostatic precipitator | 90.9 | 91.5 |
| β-caryophyllene 4 | 0.7 | 12 | 2000-4000 | 3.5 | 14 | Electrostatic precipitator | 93.9 | 94.1 |
| β-caryophyllene 5 | 0.7 | 12 | 2000-4000 | 3.5 | 9 | Electrostatic precipitator | 93.9 | 94.1 |
| Limonene 1 | 0.07 | 30 | 80-90 | 7.0 | 24 | Single stage impactor | 95.6 | 98.7 |
| Limonene 2 | 0.07 | 30 | 80-90 | 7.0 | 24 | Single stage impactor | 97.4 | 98.8 |
| Limonene 3 | 2.0 | 13 | 7000 | 3.5 | 20 | Electrostatic precipitator | 95.6 | 95.3 |
| Limonene 4 | 2.0 | 13 | 7000 | 3.5 | 20 | Electrostatic precipitator | 92.7 | 94.5 |




Table 2. Experimental conditions for production and collection of SOM produced by photo-
oxidation and measured separation relative humidities (SRH) upon moistening and merging
relative humidities (MRH) upon drying of the collected particles. Particles were collected on
hydrophobic substrates using an electrostatic precipitator or single stage impactor. SRH = 0 and
MRH = 0 indicates LLPS was not observed during humidity cycles.

| SOM sample | VOC conc. (ppm) | $O_3$ conc. (ppm) | SOM mass conc. ($\mu g\ m^{-3}$) | Flow rate for SOM particle production ($L\ m^{-1}$) | Collection time (hour) | Collection method | MRH (%) | SRH (%) |
|---|---|---|---|---|---|---|---|---|
| Toluene 1 | 0.2 | 30 | 80-100 | 7.0 | 20 | Single stage impactor | 0 | 0 |
| Toluene 2 | 0.2 | 30 | 80-100 | 7.0 | 24 | Single stage impactor | 0 | 0 |
| Toluene 3 | 1.0 | 30 | 600-1000 | 7.0 | 48 | Electrostatic precipitator | 0 | 0 |
| Toluene 4 | 1.0 | 30 | 600-1000 | 7.0 | 48 | Electrostatic precipitator | 0 | 0 |
| Toluene 5 | 1.0 | 30 | 600-1000 | 7.0 | 96 | Electrostatic precipitator | 0 | 0 |
| Toluene 6 | 1.0 | 30 | 600-1000 | 7.0 | 96 | Electrostatic precipitator | 0 | 0 |



Table 3. Summary of the LLPS results as well as the oxygen-to-carbon atomic ratios (O:C) of the
studied SOM particles. The standard deviation (σ) of the separation relative humidity (SRH) and
merging relative humdities (MRH) is derived from several cycles of RH for different SOM mass
concentrations. SRH = 0 and MRH = 0 indicates phase separation was absent in particles.

| SOM | O:C range of SOM | | Average RH (%) ± σ | |
|---|---|---|---|---|
| | Lowest | Highest | MRH | SRH |
| Ozonolysis of β-caryophyllene | 0.25[a] | 0.45[a] | 93.3 ± 1.7 | 93.8 ± 1.3 |
| Ozonnolysis of α-pinene | 0.27[b] | 0.60[c] | 95.9 ± 0.8[d] | 96.1 ± 0.6[d] |
| Ozonolysis of limonene | 0.34[e] | 0.47[f] | 95.3 ± 1.9 | 96.8 ± 2.2 |
| Photo-oxidation of isoprene | 0.52[g] | 0.85[g] | 0[h] | 0[h] |
| Photo-oxidation of toluene | 0.68[i] | 1.32[g] | 0 | 0 |

[a] Chen et al. (2011); [b] Aiken et al. (2008); [c] Reinhardt et al. (2007); [d] Renbaum-Wolff et al. (2016);
[e] Heaton et al. (2007); [f] Bateman et al. (2009); [g] Lambe et al. (2015); [h] Rastak et al. (2017); [i] Chhabra
et al. (2011)



Table 4. Literature data of measured O:C ratio, $\kappa_{HGF}$, $\kappa_{CCN}$, and the difference between $\kappa_{HGF}$ and
$\kappa_{CCN}$, denoted as $\Delta\kappa$, of SOMs.

| SOM | O:C | $\kappa_{HGF}$ at 90% RH | $\kappa_{CCN}$ | $\Delta\kappa$ | Reference |
|---|---|---|---|---|---|
| Photo-oxidation of α-pinene | 0.4 | 0.04 | 0.15 | 0.11 | Massoli et al. (2010) |
| | 0.43 | 0.07 | 0.16 | 0.09 | Massoli et al. (2010) |
| | 0.45 | 0.03 | 0.11 | 0.08 | Pajunoja et al. (2015) |
| | 0.55 | 0.10 | 0.12 | 0.02 | Pajunoja et al. (2015) |
| | 0.67 | 0.14 | 0.18 | 0.04 | Massoli et al. (2010) |
| | 0.70 | 0.12 | 0.13 | 0.01 | Pajunoja et al. (2015) |
| Photo-oxidation of isoprene | 0.86 | 0.13 | 0.14 | 0.01 | Pajunoja et al. (2015) |
| Photo-oxidation of longifolene | 0.39 | 0.02 | 0.10 | 0.08 | Pajunoja et al. (2015) |
| | 0.56 | 0.03 | 0.09 | 0.06 | Pajunoja et al. (2015) |
| | 0.83 | 0.08 | 0.10 | 0.02 | Pajunoja et al. (2015) |





Figure 1. Optical images of SOM particles with increasing RH: (a) β-caryophyllene-derived SOM for the mass concentrations of 2000 - 4000 µg m$^{-3}$ (β-caryophyllene 3, Table 1) and (b) limonene-derived SOM for the mass concentrations of 7000 µg m$^{-3}$ (Limonene 3, Table 1). Note that the light gray circles at the center of the particles are an optical effect due to the hemispherical nature of the particles. Illustrations are shown below the images for clarity. Green: SOM-rich phase. Blue: water-rich phase. The scale bar is 20 µm.





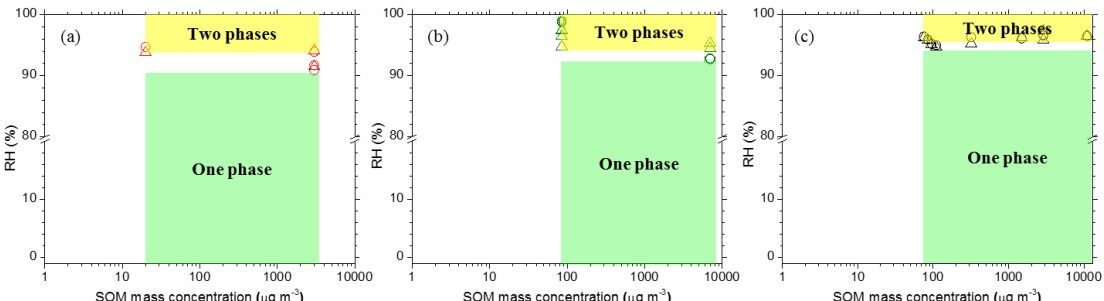

Figure 2. RH at which two phases were observed during humidity cycles of individual particles of
(a) β-caryophyllene-derived SOM and (b) limonene-derived SOM from this study, and (c) α-
pinene-derived SOM from Renbaum-Wolff et al. (2016) as a function of the SOM mass
concentrations. For all panels, circles represent onset of phase separation upon moistening (i.e.
separation relative humidity, SRH) and triangles represent merging of two liquid phases upon
drying (i.e. merging relative humidity, MRH). Yellow shaded region indicates two phases present
and green shaded region indicates one phase prevalent in SOM.





Figure 3. Optical images of toluene-derived SOM for the particle mass concentrations of 80 - 100 µg m⁻³ (Toluene 2, Table 2) with increasing RH. Illustrations of the images are shown for clarity. Green: SOM-rich phase. Size bar is 20 µm.



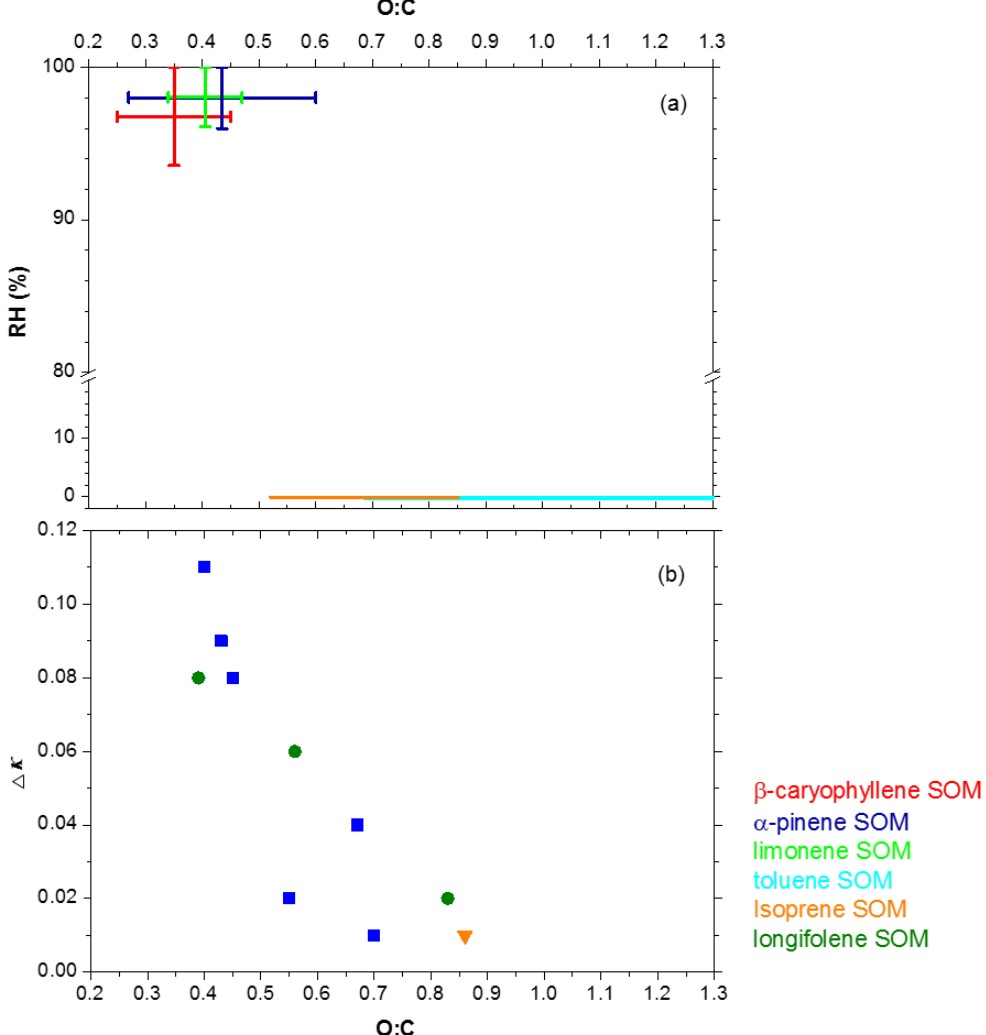

Figure 4. (a) Relative humidity in two phases as a function of the average O:C of the organic
material. Shown are the combined results for both increasing and decreasing relative humidity at
the different SOM mass concentrations from Table 3. β-caryophyllene-derived SOM (red),
limonene-derived SOM (light green), and toluene-derived SOM (cyan) from this study, isoprene-
derived SOM (orange) from Rastak et al. (2017) and α-pinene-derived SOM (blue) from Renbaum-
Wolff et al. (2016) as a function of O:C. RH value of 0 % indicates that LLPS did not occur. The





1    O:C values of the SOM particles were taken from Table 3. (b) The difference between $\kappa_{HGF}$ and

2    $\kappa_{CCN}$, denoted as $\Delta\kappa$, as a function of the average O:C of the SOM. Data taken from Table 4.

