# Peer review of "Liquid-liquid phase separation in particles containing secondary organic material free of inorganic salts"

_Atmospheric Chemistry and Physics, 2017_

## Referee Comment (RC1) · Anonymous Referee #1 · 8 Jun 2017

Summary:
I have reviewed the paper "Liquid-liquid phase separation in particles containing secondary organic material free of inorganic salts" by the authors M. Song, P. Liu, S. T. Martin, and A. K. Bertram, which was submitted to *Atmos. Chem. Phys. Discuss*. This paper expands on previous work in which Renbaum-Wolff et al. found that in particles composed of secondary organic material (SOM) generated from α-pinene, liquid-liquid phase separation (LLPS) could occur at very high relative humidities (RH > 90%) in the absence of salt. Such a result may explain discontinuities in hygroscopicity observed below and above water saturation. This paper extends this research to SOM generated from β−caryophyllene, limonene, and toluene. LLPS is observed for the first two compounds, and not for the third. Overall, the data in the paper are of interest to atmospheric chemists and will make a good contribution to this journal. I have several major and minor comments, as indicated below, which should be addressed prior to publication. Major comments are preceded by a "*".

Abstract:
* O:C ranges overlap, which could be confusing to readers. Since only three systems are studied, it would be more accurate to state that you observed phase separation for BVOC oxidation products and not for toluene SOA. Plus, the O:C ratios obtained are averages over all SOA compounds formed in the experiment. LLPS may be observed due to a small concentration of compounds at very low O:C. To state a range may therefore be misleading.

Intro:
pg 2 line 8: Both low volatility and semivolatiles can partition to the particle phase

pg 2 line 24: LLPS can occur at O:C < 0.56 and does not occur at O:C > 0.8 according to your previous papers

pg 2 line 27: Peters is not cited. Perhaps you mean Petters et al.?

pg 2-3 lines 29,30,1: Change to "which could result in altered CCN properties" or similar, as the results from Renbaum-Wolff et al. are simulations of CCN data rather than experiments

pg 3 lines 1-4: The results of Hodas et al. ACP 2016, 16, 12767-12792 should be discussed here.

Materials and Methods:
The error for the hygrometer (2.0% RH) seems high. Can the authors comment further about where this error is coming from (e.g. rate of RH ramp)?

Results and Discussion:
* pg 6: More evidence is needed to support the claim that LLPS is occurring by spinodal decomposition. Comparing to the data in Ciobanu et al. 2009, both the results in this paper and Renbaum-Wolff et al. 2016 clearly look like nucleation and growth (formation of small inclusions), as no schlieren are observed.

* pg 6: It is reasonable to assume that the water-rich organic component is in the core of the particle and the more-insoluble organic is in the shell, but what evidence is there for this assignment?

* pg 6: It is important to note that while core-shell structures are observed for these systems, without the substrate, they could be either partially engulfed or core-shell. It is less clear that there would be an effect of LLPS on CCN for these systems if they are partially engulfed in morphology.

pg 7 line 4: The term "mixing" is more consistent with previous literature in this field (see for example Fig. 2 in the review from your group by You et al. in 2014). Later in this section, "MRH" is the mixing RH.

pg 7 line 8: Do you expect that particle mass concentrations will change O:C? Why is there a question that this will have an effect on LLPS?

pg 8 lines 15-23: SOA has a range of O:C values; the best that can be reported are average values. It could just be the very low O:C compounds that phase separate. It is therefore hard to understand how the systems used in this experiment can be compared to systems composed of only three components.

Implications:
pg 9 lines 9-13: The Hodas et al. reference mentioned above should be added here.

Table 1&2: It would be nice to list average O:C for these exact systems, if known.

* Table 3: This table shows the minimum and maximum O:C range observed in the literature for SOM derived from these species, rather than O:C from the experiment run in this study. This is not ideal, but it may be due to experimental limitations. I worry, however, that O:C will change dramatically depending on how a system is oxidized and for how long. Are these comparable techniques to what you used? Do the numbers represent both chamber and flow tube studies? More information is needed here or in the text to understand how to interpret this data. Plus, Table 3 & 4 are not consistent with one another (the highest O:C for alpha-pinene is 0.70 according to Table 4).

Figure 4: The first sentence of the legend is not clear. You are plotting SRH and MRH for the systems.

---

## Referee Comment (RC2) · Anonymous Referee #2 · 12 Jun 2017

Summary. Previously, it was found that at high RH (>95%), $\alpha$-pinene-derived SOM particles free of inorganic salts can undergo LLPS, while isoprene-derived SOM particles free of inorganic salts do not. In this paper, additional SOM particles free of inorganic salts where studied. It was found that SOM generated from ozonolysis of $\beta$-caryophyllene and limonene that are similar to $\alpha$-pinene-derived SOM, while particles generated by photo-oxidation of toluene that are similar to isoprene-derived SOM. In addition to the LLPS information, the authors found a relationship between occurrence of LLPS and the average oxygen-to-carbon elemental ratio (O:C) of the organic material. Low O:C ratio resulted in LLPS. Publication is recommended with minor revisions.

[Figure]

Concerns.

On page 5, the authors mention that 20 – 80 micron diameter particles are required for LLPS. Is there a reference to that? Why is there a size dependence in the observations? In addition, is there a RH rate of change dependence? More discussion is needed.

While the observation of LLPS or no LLPS with SOM derived in various ways is interesting and important to the atmospheric chemistry community, the manuscript would be stronger with proposed explanations or discussions on *why*. Discussion on differences in $\alpha$-pinene-derived SOM and isoprene-derived SOM structures, or the structures of the components studied here might have been helpful.

In Table 1, some MRH values are higher than the SRH values. This seems counter-intuitive. Discussion is needed justifying these observations.

How are the SOM mass concentrations at phase chance calculated (figure 2, x-axis)? Is the volume of the droplet known? What is the change in contact angle of the droplet with the surface, as a function of RH?

Typo.

Page 8 – 2nd line – "observed". It should be "observe".

---

## Author Comment (AC1) · 8 Aug 2017

David Topping, Co-Editor of ACP

Dear David Topping,

Listed below are our responses to the comments from the reviewers of our manuscript. We thank the reviewers for carefully reading our manuscript and for their very helpful suggestions! For clarity and visual distinction, the referee comments or questions are listed here in black and are preceded by bracketed, italicized numbers (e.g. [1]). Authors' responses are in red below each referee statement with matching numbers (e.g.

[Figure]

[A1]).

Sincerely,

Allan Bertram Professor of Chemistry University of British Columbia

Response to Referee #1 (Reviewer comments in black text)

Summary: I have reviewed the paper "Liquid-liquid phase separation in particles containing secondary organic material free of inorganic salts" by the authors M. Song, P. Liu, S. T. Martin, and A. K. Bertram, which was submitted to Atmos. Chem. Phys. Discuss. This paper expands on previous work in which Renbaum-Wolff et al. found that in particles composed of secondary organic material (SOM) generated from a-pinene, liquid-liquid phase separation (LLPS) could occur at very high relative humidities (RH > 90%) in the absence of salt. Such a result may explain discontinuities in hygroscopicity observed below and above water saturation. This paper extends this research to SOM generated from b-caryophyllene, limonene, and toluene. LLPS is observed for the first two compounds, and not for the third. Overall, the data in the paper are of interest to atmospheric chemists and will make a good contribution to this journal. I have several major and minor comments, as indicated below, which should be addressed prior to publication.

Major comments are preceded by a "*".

[1] Abstract: * O:C ranges overlap, which could be confusing to readers. Since only three systems are studied, it would be more accurate to state that you observed phase separation for BVOC oxidation products and not for toluene SOA. Plus, the O:C ratios obtained are averages over all SOA compounds formed in the experiment. LLPS may be observed due to a small concentration of compounds at very low O:C. To state a range may therefore be misleading.

[A1] Thank you for the comment. To avoid confusion, in the Abstract we will remove the sentence that mentions the O:C range. In addition, we will remove "average" from

the Abstract since LLPS can depend on both the average O:C and distribution of O:C.

Intro: [2] pg 2 line 8: Both low volatility and semivolatiles can partition to the particle phase

[A2] We will adjust this sentence to make it clear that semivolatiles can also partition to the particle phase.

[3] pg 2 line 24: LLPS can occur at O:C < 0.56 and does not occur at O:C > 0.8 according to your previous papers

[A3] The LLPS range will be corrected in the revised manuscript to make this point clear.

[4] pg 2 line 27: Peters is not cited. Perhaps you mean Petters et al.?

[A4] Correct! This will be revised on in the final version.

[5] pg 2-3 lines 29,30,1: Change to "which could result in altered CCN properties" or similar, as the results from Renbaum-Wolff et al. are simulations of CCN data rather than experiments

[A5] We will revise as suggested.

[6] pg 3 lines 1-4: The results of Hodas et al. ACP 2016, 16, 12767-12792 should be discussed here.

[A6] We will included the reference and discussion in the revised manuscript.

[7] Materials and Methods: The error for the hygrometer (2.0% RH) seems high. Can the authors comment further about where this error is coming from (e.g. rate of RH ramp)?

[A7] The RH was determined from measurements of the temperature with a thermocouple and measurements of the dew point/frost point with a chilled mirror sensor (General Eastern, Canada). The RH was calibrated using the deliquescence RH for pure

ammonium sulfate particles (80% RH at 293 K, Martin, 2000). After calibration, the uncertainty of the hygrometer was ± 2.0 % RH based on the reproducibility of multiple deliquesce measurements. For clarification, this information will be added to the revised manuscript.

[8] Results and Discussion: * pg 6: More evidence is needed to support the claim that LLPS is occurring by spinodal decomposition. Comparing to the data in Ciobanu et al. 2009, both the results in this paper and Renbaum-Wolff et al. 2016 clearly look like nucleation and growth (formation of small inclusions), as no schlieren are observed.

[A8] Spinodal decomposition is only clear in the movies provided in the Supplementary Material of this manuscript. Since this is not an important point in our paper, we will remove the two sentences from the paper that refer to spinodal decomposition.

[9] * pg 6: It is reasonable to assume that the water-rich organic component is in the core of the particle and the more-insoluble organic is in the shell, but what evidence is there for this assignment?

[A9] We assume the core of the particle is water-rich, in part, because the size of the core decreases and eventually disappears as the RH decreases. The surface tension of water and the surface tensions of more-oxidized and less-oxidized organics are consistent with this assumption (Jasper, 1972). To address the referee's comment, we will add this information to the revised manuscript.

Jasper, J. J.: The surface tension of pure liquid compounds, J. Phys. And Chem. Ref. Data, vol 1, 841-1009, Doi: http://dx.doi.org/10.1063/1.3253106, 1972.

[10] * pg 6: It is important to note that while core-shell structures are observed for these systems, without the substrate, they could be either partially engulfed or core-shell. It is less clear that there would be an effect of LLPS on CCN for these systems if they are partially engulfed in morphology.

[A10] We agree that a different morphology could result if the substrate is removed.

We will add this point to the manuscript to address the referee's comment.

[11] pg 7 line 4: The term "mixing" is more consistent with previous literature in this field (see for example Fig. 2 in the review from your group by You et al. in 2014). Later in this section, "MRH" is the mixing RH.

[A11] In the revised manuscript, we will replace "merging" with "mixing" as suggested.

[12] pg 7 line 8: Do you expect that particle mass concentrations will change O:C? Why is there a question that this will have an effect on LLPS?

[A12] Previous work has shown that the O:C of SOM can depend on the particle mass concentrations used to generate the SOM (E.g. Shilling et al. ACP, 2009, 771-782). To address this the referee's comments, this information will be added to the revised manuscript.

[13] pg 8 lines 15-23: SOA has a range of O:C values; the best that can be reported are average values. It could just be the very low O:C compounds that phase separate. It is therefore hard to understand how the systems used in this experiment can be compared to systems composed of only three components.

[A13] We completely agree that the spread of O:C values in the mixture is important, as well as the average O:C values. We will add the text below to the revised manuscript to make this clear.

"SOM have an average O:C and a spread (or distribution) of O:C values. Similar to SOM, systems containing two organics and water also have a spread in O:C and an average O:C. Hence, as a starting point to understanding LLPS in SOM, we considered previous studies that explored the miscibility gap in bulk solutions containing two organics and water (see Table 1 in Ganbavale et al., 2015)."

[14] Implications: pg 9 lines 9-13: The Hodas et al. reference mentioned above should be added here.

[A14] This reference will be included in the revised manuscript.

[15] Table 1&2: It would be nice to list average O:C for these exact systems, if known.

[A15] We did measure the O:C of the toluene-derived SOM studied in our manuscript. This information will be added to the Methods in the revised manuscript. The O:C of the other SOM was not measured by us; hence we rely on average O:C measurements from other studies using similar conditions. See [A16] below.

[16] * Table 3: This table shows the minimum and maximum O:C range observed in the literature for SOM derived from these species, rather than O:C from the experiment run in this study. This is not ideal, but it may be due to experimental limitations. I worry, however, that O:C will change dramatically depending on how a system is oxidized and for how long. Are these comparable techniques to what you used? Do the numbers represent both chamber and flow tube studies? More information is needed here or in the text to understand how to interpret this data. Plus , table 3 & 4 are not consistent with one another (the highest O:C for alpha-pinene is 0.70 according to Table 4).

[A16] Regarding Table 3: The O:C of toluene-derived SOM investigated in this study was determined by us. This information will be added to the revised manuscript. The O:C of the other SOM investigated in our manuscript were not measured by us; hence, we rely on the O:C measurements from other studies. In the initial version of the manuscript, we included all previous O:C measurements. The referee is correct that O:C can change based on oxidation level and oxidation time. To address the referee's comments, in Table 3, we will only include O:C values from the literature that were determined using similar experimental conditions to those used in the LLPS studies. To illustrate the similarity in experimental conditions, in the revised manuscript (Table S1) we will list the experiment conditions used to generate the SOM in the LLPS studies and the experimental conditions used to generate the SOM in the O:C studies reported in Table 3.

Regarding Table 4: The average O:C values reported in Table 4 are based on measurements in from the individual studies. The experimental conditions for the studies reported in Table 4 are not necessarily similar to the experimental conditions for the studies reported in Table 3, even if the same precursor volatile organic compound was used. For clarity, this information will be added to the revised manuscript.

[17] Figure 4: The first sentence of the legend is not clear. You are plotting SRH and MRH for the systems.

[A17] We will revise the sentence for clarify.

Please also note the supplement to this comment:
https://www.atmos-chem-phys-discuss.net/acp-2017-408/acp-2017-408-AC1-supplement.pdf

**Supplement:**

*Supplementary Material of*

**Liquid-liquid phase separation in particles containing secondary organic material free of inorganic salts**

**M. Song et al.**

Correspondence to: A. K. Bertram (bertram@chem.ubc.ca)

Section S1. The average O:C values of the SOM used in the LLPS studies (excepted for toluene-derived SOM) were based on average O:C values reported in the literature (Table 3). Since the average O:C can depend on oxidant time and oxidation conditions, we chose O:C values from the literature that were determined using similar experimental conditions to those used in the LLPS studies. To illustrate the similarity in experimental conditions, in Table S1 we list the experimental conditions used to generate the SOM in the LLPS studies and the experimental conditions used to generate the SOM in the O:C studies referenced in Table 3.

Table S1. Experimental conditions used to generate SOM for the liquid-liquid phase separation (LLPS) studies as well as the experimental conditions used to generate the SOM in the O:C studies referenced in Table 3. '*NA*' indicates not available.

| SOM | VOC conc. (ppm) | $O_3$ conc. (ppm) | SOM generation | Residence time (sec.) | O:C | Type of study |
|---|---|---|---|---|---|---|
| Ozonolysis of β-caryophyllene | 0.03 - 0.7 | 12 - 30 | Flow tube reactor | 38 | *NA* | LLPS (This study) |
| | 0.1 | 15 | Oxidation flow reactor | 110 | 0.36 – 0.38 | O:C analysis (Li et al., 2015) |
| Ozonolysis of α-pinene | 0.2 – 5.0 | 10 - 20 | Flow tube reactor | 38 | *NA* | LLPS (Renbaum-Wolff et al., 2016) |
| | 0.1 | 15 | Oxidation flow reactor | 110 | 0.42 - 0.44 | O:C analysis (Li et al., 2015) |
| Ozonolysis of limonene | 0.07 – 2.0 | 13 - 30 | Flow tube reactor | 38 | *NA* | LLPS (This study) |

| | | | | | | |
|---|---|---|---|---|---|---|
| | 41 | 1 | Flow tube reactor | 110 | 0.34 - 0.40 | O:C analysis (Heaton et al., 2007) |
| Photo-oxidation of isoprene | 0.7 – 7.0 | 10 - 30 | Oxidation flow reactor | 84 -114 | *NA* | LLPS (Rastak et al., 2016) |
| | 0.7 | 15 | Oxidation flow reactor | 110 | 0.87 – 0.89 | O:C analysis (Li et al., 2015) |
| | 0.33 | 15 - 30 | Oxidation flow reactor | 100 | 0.52 – 0.85 | O:C analysis (Lamb et al., 2015) |

**References**

Chen, Q., Liu, Y., Donahue, N. M., Shilling, J. E. & Martin, S. T. Particle-phase chemistry of secondary organic material: modeled compared to measured O:C and H:C elemental ratios provide constraints. Environ. Sci. Technol. 45, 4763-4770, doi:10.1021/es104398s, 2011.

Heaton, K. J., Dreyfus, M. A., Wang, S., and Johnston, M. V.: Oligomers in the early stage of biogenic secondary organic aerosol formation and growth, Environ. Sci. Technol., 41, 6129-6136, 10.1021/es070314n, 2007.

Lambe, A. T., Chhabra, P. S., Onasch, T. B., Brune, W. H., Hunter, J. F., Kroll, J. H., Cummings, M. J., Brogan, J. F., Parmar, Y., Worsnop, D. R., Kolb, C. E., and Davidovits, P.: Effect of oxidant concentration, exposure time, and seed particles on secondary organic aerosol chemical composition and yield, Atmos. Chem. Phys., 15, 3063-3075, 10.5194/acp-15-3063-2015, 2015.

Li, Y. J., Liu, P. F., Gong, Z. H., Wang, Y., Bateman, A. P., Bergoend, C., Bertram, A. K., and Martin, S. T.: Chemical Reactivity and Liquid/Nonliquid States of Secondary Organic Material, Environ Sci Technol, 49, 13264-13274, 10.1021/acs.est.5b03392, 2015.

Rastak, N., A. Pajunoja, J. C. Acosta Navarro, D. G. Partridge, J. Ma, M. Song, A. Kirkevåg, Y. Leong, W. W. Hu, N. F. Taylor, D. R. Collins, K. Cerully, A. Bougagioti, R. Krejci, P. Liu, T. Petäjä, C. Percival, A. M. L. Ekman, A. Nenes, S. T. Martin, J. L. Jimenez, D. O. Topping, A. K. Bertram,  A. Zuend, A. Virtanen, and I. Riipinen: Microphysical explanation of the RH-dependent water-affinity of biogenic organic aerosol and its importance for climate, Geophys. Res. Lett., accepted, 2017.

Renbaum-Wolff, L., Song, M. J., Marcolli, C., Zhang, Y., Liu, P. F. F., Grayson, J. W., Geiger,

F. M., Martin, S. T., and Bertram, A. K.: Observations and implications of liquid-liquid phase separation at high relative humidities in secondary organic material produced by alpha-pinene ozonolysis without inorganic salts, Atmos. Chem. Phys., 16, 7969-7979, 10.5194/acp-16-7969-2016, 2016.

---

## Author Comment (AC2) · 8 Aug 2017

David Topping, Co-Editor of ACP

Dear David Topping,

Listed below are our responses to the comments from the reviewers of our manuscript. We thank the reviewers for carefully reading our manuscript and for their very helpful suggestions! For clarity and visual distinction, the referee comments or questions are listed here in black and are preceded by bracketed, italicized numbers (e.g. [1]). Authors' responses are in red below each referee statement with matching numbers (e.g.

[Figure]

[A1]).

Sincerely,

Allan Bertram Professor of Chemistry University of British Columbia

Response to Referee #2 (Reviewer comments in black text)

Summary. Previously, it was found that at high RH (>95%), $\alpha$-pinene-derived SOM particles free of inorganic salts can undergo LLPS, while isoprene-derived SOM particles free of inorganic salts do not. In this paper, additional SOM particles free of inorganic salts where studied. It was found that SOM generated from ozonolysis of caryophyllene and limonene that are similar to $\alpha$-pinene-derived SOM, while particles generated by photo-oxidation of toluene that are similar to isoprene-derived SOM. In addition to the LLPS information, the authors found a relationship between occurrence of LLPS and the average oxygen-to-carbon elemental ratio (O:C) of the organic material. Low O:C ratio resulted in LLPS. Publication is recommended with minor revisions. Discussion paper Concerns.

[1] On page 5, the authors mention that 20 – 80 micron diameter particles are required for LLPS. Is there a reference to that? Why is there a size dependence in the observations?

[A1] The resolution of the microscope used in the current experiments was roughly 1 micron. From experience, detection of LLPS with our microscope setup is the clearest when the size of the particles are roughly 20 - 80 micron, although smaller sizes are possible with optical microscopy. We will re-phrase the sentence referred to by the referee for clarity. We did not observe a size dependence for the LLPS phase transition for the relative narrow range of sizes investigated.

[2] In addition, is there a RH rate of change dependence? More discussion is needed.

[A2] We did not observe a dependence of LLPS on the RH ramp rate, although only a narrow range of rates were used. This information will be added to the revised

manuscript.

[3] While the observation of LLPS or no LLPS with SOM derived in various ways is interesting and important to the atmospheric chemistry community, the manuscript would be stronger with proposed explanations or discussions on *why*. Discussion on differences in $\alpha$-pinene-derived SOM and isoprene-derived SOM structures, or the structures of the components studied here might have been helpful.

[A3] We certainly agree that an explanation of "why" LLPS is observed in some cases but not others is important. Previous thermodynamic modelling studies give some explanations for why LLPS occurs in $\alpha$-pinene-derived SOM but not isoprene-derived SOM (See Renbaum-Wolff et al., 2016; Rastak et al., 2017). We hope to explore this question in more detail in the future by investigate LLPS in organic particles containing mixtures of commercially available organic compounds.

[4] In Table 1, some MRH values are higher than the SRH values. This seems counterintuitive. Discussion is needed justifying these observations.

[A4] The uncertainty in MRH and SRH values reported in Table 1 is $\pm$ 2.0 % RH, due to the uncertainty in the RH measurement. As a result, the SRH and MRH values agree within the uncertainty of the measurements. To address the referee's comment, in the caption to Table 1, we will point out that the uncertainty in the MRH and SRH is $\pm$ 2.0 % RH, due to the uncertainty in the RH measurements.

[5] How are the SOM mass concentrations at phase chance calculated (figure 2, x-axis)?

[A5] The mass concentrations of SOM were determined from measurements of the number-diameter distribution of SOM particles in the flow tube reactor or OFR. This information will be added to manuscript for clarity.

[6] Is the volume of the droplet known? What is the change in contact angle of the droplet with the surface, as a function of RH?

[A6] The contact angle of the droplets with the surface was not measured in our experiments, and hence an accurate volume of the droplets was not known. From the optical images, we only determined the projected diameter of the droplets.

[7] Typo. Page 8 – 2nd line – "observed". It should be "observe". [A7] This change will be made in the revised manuscript.

———————————————